# Testing the mean field theory of scalar field dark matter

Andrew Eberhardt[1,2,3*], Alvaro Zamora[1,2,3], Michael Kopp,[4] and Tom Abel[1,2,3]

**1** Kavli Institute for Particle Astrophysics and Cosmology, Menlo Park, 94025, California, USA
**2** Physics Department, Stanford University, Stanford, California, USA
**3** SLAC National Accelerator Laboratory
**4** Stockholm University, Hannes Alfvéns väg 12, SE-106 91 Stockholm, Sweden

★ aeberhar@stanford.edu

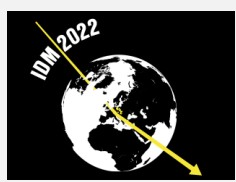 *14th International Conference on Identification of Dark Matter*
## Abstract

Scalar field dark matter offers an interesting alternative to the traditional WIMP dark matter picture. Astrophysical and cosmological simulations are useful to constraining the mass of the dark matter particle in this model. This is particularly true at low mass where the wavelike nature of the dark matter particle manifests on astrophysical scales. These simulations typical use a classical field approximation. In this work, we look at extending these simulations to include quantum corrections. We look into both the ways in which large corrections impact the predictions of scalar field dark matter, and the timescales on which these corrections grow large. Corrections tend to lessen density fluctuations and increase the effect of "quantum pressure". During collapse, these corrections grow exponentially, quantum corrections would become important in about $\sim 30$ dynamical times. This implies that the predictions of classical field simulations may differ from those with quantum corrections for systems with short dynamical times.

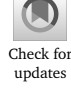
## 1 Introduction

Scalar field dark matter is an interesting model as it exhibits wave-like properties on scales of the deBroglie wavelength, [1–3]. On the low mass end, $m \sim 10^{-22} - 10^{-19}$ eV wavelike phenomena manifest on astrophysical scales. This alters the structure associated with cold dark matter by creating $\mathcal{O}(1)$ density fluctuations and "quantum pressure" which wash out structure on small scales. Simulations of structure formation, therefore, provide a powerful tool to constrain the lower mass bound of the dark matter [4–7]. Typically these simulations are performed using the classical equations of motion, the Schödinger-Poisson equations.

For free fields at high occupations, like those typical of electromagnetism, the mean field theory is known to accurately describe the evolution of observables [8]. However, when non-linear interactions are present, such as gravity, the classical field theory can admit quantum corrections on some time scale [9–12]. In this work, we consider the implications of quantum corrections for the predictions of classical field theory simulations. We accomplish this by simulating the leading order quantum corrections to the classical equations of motion, described in [9, 10, 12–14]. We find that large scale quantum corrections tend to remove phase information from the field and wash out $\mathcal{O}(1)$ density fluctuations. Corrections grow exponentially during nonlinear collapse resulting in a logarithmic enhancement in the breaktime with number of particles, i.e. $t_{br} \propto \log(n_{tot})$. This is implies that even large systems, such as dark matter halos of galaxies, which have dynamical times short compared to the age of the Universe may exhibit quantum corrections altering predictions of the classical theory.

## 2 Numerical methods

### 2.1 Mean field theory

There are many methods outlined in the literature for simulating the classical field equations, for example [2, 15]. A single complex non-relativistic scalar field is defined, where the square amplitude corresponds to the spatial density, i.e. $\psi(x) = \sqrt{\rho(x)}e^{i\phi(x)}$, and the phase gradient to the bulk velocity, when expressed in the position basis, i.e. $\hbar\nabla\phi(x) = v(x)$.

We then compute the evolution using the Schrödinger-Poisson equations

$$\partial_t \psi(x) = -i\left(\frac{-\tilde{\hbar}\nabla^2}{2} + \frac{V(x)}{\tilde{\hbar}}\right)\psi(x), \tag{1}$$

$$\nabla^2 V(x) = 4\pi G \rho(x), \tag{2}$$

where $\tilde{\hbar} \equiv \hbar/m$, and $m$ is the mass of the field. These equations are integrated using the single classical field scheme described in [15].

### 2.2 Field moment expansion

For systems of about $M = 256$ spatial modes and $n_{tot} < 10^{10}$ total particles, direct integration of Schrödinger's equation is not feasible as the relevant Hilbert space is far too large, i.e. $\mathcal{D}[\mathcal{H}] \sim 10^{2000}$. Instead we simulate the mean field theory plus the leading order quantum correction proportional to the width of the underlying quantum distribution in field space. The method and solver we use is detailed in [13]. Like in the mean field case we start with a classical field, $\psi(x, t)$. In addition to this field we also track a field variance and covariance, $\langle\delta\psi(x)\delta\psi(y)\rangle$ and $\langle\delta\psi^\dagger(x)\delta\psi(y)\rangle$ respectively.

### 2.3 Truncated Wigner method

It is also possible to approximate the evolution of the Wigner function using an ensemble of classical fields. This method is described in [14]. A quantum state has a pseudo phase space representation given by the Weyl symbol of the density matrix, $\hat{\rho}$,

$$W[\psi(x), \psi^*(x)] = \frac{1}{\text{Norm}} \int\int d\eta^* d\eta \langle\psi(x) - \frac{\eta}{2}|\hat{\rho}|\psi(x) + \frac{\eta}{2}\rangle \times \tag{3}$$

$$e^{-|\psi(x)|^2 - \frac{|\eta|^2}{4}} e^{\frac{1}{2}(\eta^*\psi(x) - \eta\psi^*(x))},$$

for a pure state $\hat{\rho} = |\phi\rangle\langle\phi|$. Where $|\phi\rangle$ is some quantum state. If $|\phi\rangle$ is a coherent state then the corresponding Wigner function is Gaussian, i.e. $W[\psi(x), \psi^*(x)] = \frac{1}{\pi}e^{-|\psi(x)-\psi^{cl}(x)|^2}$.

We can approximate this distribution as an ensemble of $N_s$ classical fields, $\Psi = \{\psi_1, \psi_2, \ldots, \psi_{N_s}\}$, in analogy with corpuscular solver approximations of classical phase space. The ensemble constituents, $\psi_i(x) \sim W[\psi(x), \psi^*(x)]$, are drawn randomly from the Wigner function at each point, $x$, in our spatial grid. In the large $N_s$ limit the statistics of this ensemble can be used to approximate operators. For the symmetrically ordered operator $\hat{\Omega}[\hat{\psi}(x), \hat{\psi}^\dagger(x)]$, with Weyl symbol $\Omega_W[\psi(x), \psi^*(x)]$, we can write $\langle\hat{\Omega}[\hat{\psi}(x), \hat{\psi}^\dagger(x)]\rangle \approx \frac{1}{N_s}\sum_i \Omega_W(\psi_i, \psi_i^*)$. The Weyl symbol of a symmetrically ordered operator is simply constructed by substituting $\hat{\psi} \to \psi$.

The time evolution of the density matrix is given by the von Neumann equation $i\hbar\,\partial_t\hat{\rho} = [\hat{H}, \hat{\rho}]$. The Weyl symbol of the commutator is the Moyal bracket $\{\{\ldots\}\}_M = 2\sinh\left[\frac{1}{2}\{\ldots\}_c\right]$, where $\{\ldots\}_c$ is the classical Poisson bracket. In the limit $|\psi|^2 \gg 1$ we can approximate the Moyal braket as a Poisson bracket. We can now write the evolution of the density matrix as

$$\partial_t W[\psi(x), \psi^*(x)] = \frac{-i}{\hbar}\{\{H_W, W\}\}_M \approx \frac{-i}{\hbar}\{H_W, W\}_c \approx \frac{-i}{\hbar}\sum_i\{H_W, \psi_i\}_c, \qquad (4)$$

where $H_W$ is the Weyl symbol of the Hamiltonian. The first inequality is valid when $|\psi|^2 \gg 1$ and the second when $N_s \gg 1$. Note that $\frac{-i}{\hbar}\{H_W, \psi\}$ is just the right-hand side of equation (1), the classical field equation of motion. And so we can approximate the evolution of the Wigner function as an ensemble of classical fields each, independently, obeying the Schrödinger-Poisson equations.

## 3 Results

### 3.1 Effect of large corrections

The effect of large quantum corrections gives us insight into which observables are most perturbed by quantum effects, see figure 1. We run a simulation of the gravitational collapse of an initial over density with $M = 256$, $N_s = 4096$, $\psi^{cl}(x, t = 0) = \sqrt{M_{tot}}\sqrt{1 + 0.1\cos(2\pi x/L)}/\text{Norm}$, in a single spatial dimension. In simulation units we have $4\pi G = 0.1$ and $\tilde{\hbar} = 2.5 \times 10^{-4}$. The box size, $L$, and total mass, $M_{tot}$, are normalized to unity.

We can see that during the collapse phase the field undergoes phase diffusion but that the density remains well approximated by the MFT until the collapse. Following the collapse $O(1)$ density fluctuations are smoothed out in proportion to the amount of phase diffusion achieved prior to the collapse. Without well defined phase gradients the usual interference pattern is washed out. The effect of quantum pressure is also exaggerated.

### 3.2 Time scale of correction growth

The leading order correction term is proportional to second order central difference operators, e.g. $\langle\delta\hat{\psi}(x)\delta\hat{\psi}^\dagger(x)\rangle = \langle|\hat{\psi}(x) - \langle\hat{\psi}(x)\rangle|^2\rangle$. We can compare this operator to the square field amplitude, $\sim n_{tot}$, to approximate the size of the leading order quantum correction, see [13]. We define the value parameter

$$Q(t) = \frac{1}{n_{tot}}\int dx\,\langle\delta\hat{\psi}(x, t)\delta\hat{\psi}^\dagger(x, t)\rangle \qquad (5)$$

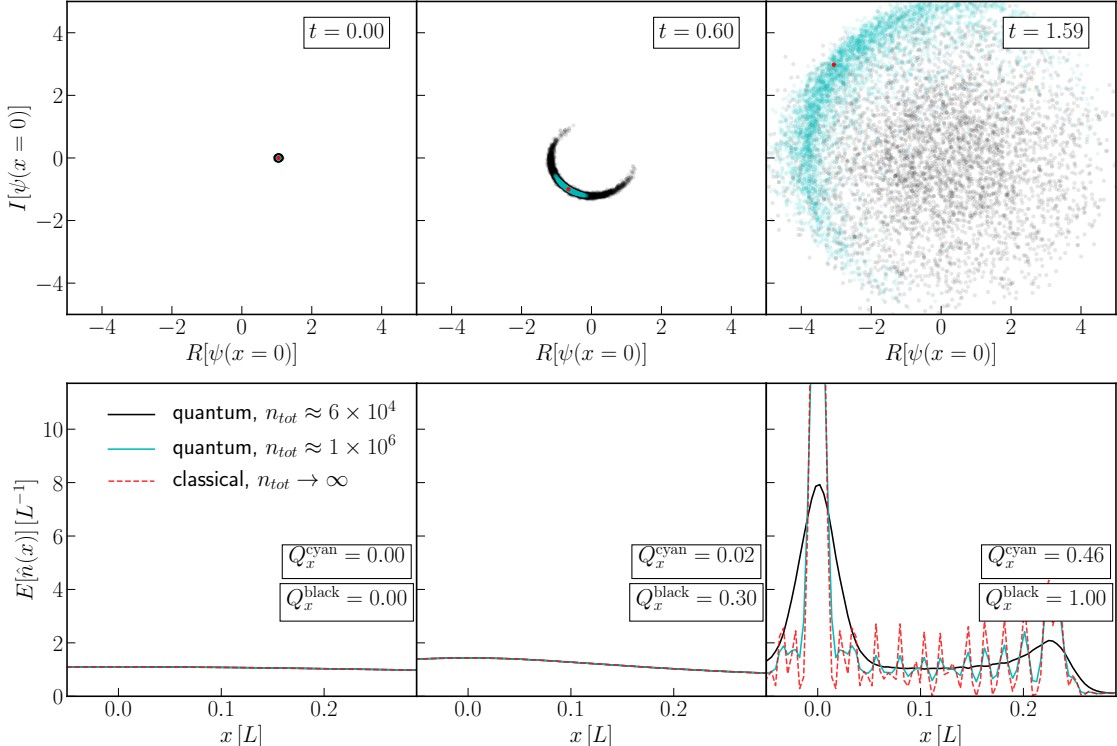

Figure 1: We plot the evolution of a spatial overdensity in one spatial dimension using the truncated Wigner expansion method. We plot the results for the classical field theory in red, and two quantum simulations with $n_{tot} \approx 6 \times 10^4$ and $n_{tot} \approx 1 \times 10^6$ in black and cyan respectively. Each column represents a different time, $t$. The top row shows the value of the each stream in the ensemble at $x = 0$ and the bottom row shows the spatial density plotted such that each field has the same norm. Shell crossing, the moment of highest density, occurs at $t = 1$.

to approximate the size of corrections. By studying how this quantity grows we can estimate how long it takes for quantum corrections to grow large.

We can see from figure 2 that $Q$ grows in a number of stages, see [10]. Initially $Q(t)$ grows quadratically. During the nonlinear collapse of the overdensity, $Q(t)$ grows exponentially as $Q(t) \sim e^{7t/t_d}$. Following the collapse, $Q(t)$ slows to a powerlaw growth. If we assume this behavior holds for a realistic system that is constantly undergoing collapse and merging events, this would imply that $Q(t) \sim 1$ happens when $e^{7t} \sim n_{tot}$, at such high occupations this time depends only weakly on the initial quadratic growth. This would imply a breaktime

$$t_{br} \sim \frac{\ln(n_{tot})}{7} t_d \,. \tag{6}$$

This is about $\sim 30$ dynamical times at $n_{tot} \sim 10^{100}$. We therefore expect quantum corrections to manifest most quickly for chaotic continuously merging systems with short dynamical times.

## 4 Conclusion

In this paper we have looked at the effect of large quantum corrections and the timescales on which they become large by studying the behavior of second order operators during the

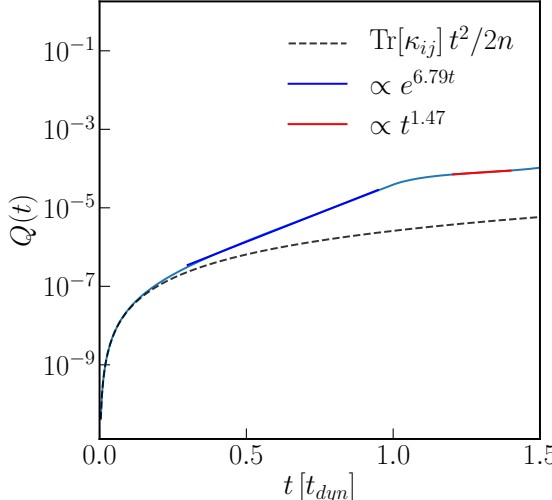

Figure 2: Here we plot the evolution of $Q(t)$ for the gravitational collapse of a spatial overdensity in one spatial dimension using the field moment expansion method. We can see that $Q$ initially grows quadratically from near 0. Then exponentially during the nonlinear growth of the overdensity, prior to the collapse, which occurs at $t = t_d$. Following the collapse we see that the growth slows to a power law. In this simulation $n_{tot} \sim 10^{10}$. $\kappa_{ij} \equiv 2\mathbb{R}\left[\sum_{kplbc} \Lambda_{pl}^{ij} \Lambda_{bc}^{kj} \langle \hat{a}_b \rangle \langle \hat{a}_c \rangle \langle \hat{a}_p^\dagger \rangle \langle \hat{a}_l^\dagger \rangle\right]$

gravitational collapse of an overdensity. We have demonstrated that the quantum corrections have the effect of removing phase information in the system which results in a smoothed density profile and exaggerated quantum pressure effect, see figure 1. Both of these effects would impact existing bounds on SFDM if present. We have also estimated the time scale on which these corrections grow. During the nonlinear growth of the collapse the corrections grow exponentially, see figure 2. We estimate that for systems with $n_{tot} \sim 10^{100}$ particles, quantum corrections become large, i.e. $Q \sim 1$, at $t \sim 30\,t_d$.

This implies a number of interest prospects for future work. Investigation of the quantum breaktime in three dimensions and how quantum corrections effect haloscope results would be of interest. Additionally, since low masses are expected to decohere rapidly an investigation of the pointer states would be useful to determine whether the classical field approximate can be applied.

## Acknowledgements

Some of the computing for this project was performed on the Sherlock cluster.

**Funding information** A.E., and T.A. are supported by the U.S. Department of Energy under contract number DE-AC02-76SF00515.

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
