# Peer review of "Testing the mean field theory of scalar field dark matter"

_SciPost Physics Proceedings, doi:SciPost Phys. Proc. 12, 058 (2023)_

## Round 1 · Referee Report · Anonymous · 2022-10-23

Strengths
1. This manuscript tries to take into account quantum corrections in fuzzy dark matter evolution, beyond the usual classical field approximation.
2. With both simulations and analytical calculations, the result suggests that there exists a non-linear growth regime where quantum corrections grow exponentially.
Weaknesses
Most of the study is done within a simplified one-dimensional framework, which may change in our real Universe.
Moreover, there are typos that need to be corrected in later editing, such as "This is implies that", "of the each stream" and ",." in Eq.(6), among others.
Report
The submission is concise, well-written. And the results are tentative, yet quite encouraging. Thus, it meets the criteria of SciPost Physics Proceedings, and should be published here.

---

## Editorial Decision

published